# Effects of Dietary Tributyrin on Growth Performance, Biochemical Indices, and Intestinal Microbiota of Yellow-Feathered Broilers

**DOI:** 10.3390/ani11123425

**Published:** 2021-12-01

**Authors:** Li Gong, Gengsheng Xiao, Liwei Zheng, Xia Yan, Qien Qi, Cui Zhu, Xin Feng, Weilong Huang, Huihua Zhang

**Affiliations:** 1School of Life Science and Engineering, Foshan University, Foshan 528000, China; gongli2018by@zju.edu.cn (L.G.); gsxiao2021@163.com (G.X.); zhenglwgkxm@163.com (L.Z.); qiqien1987@163.com (Q.Q.); juncy2010@gmail.com (C.Z.); 3haofx@163.com (X.F.); 2Laboratory of Livestock and Poultry Breeding, Institute of Animal Science, Guangdong Academy of Agricultural Sciences, Guangzhou 510000, China; xiayan1@163.com; 3Kaiping Lvhuang Agriculture and Animal Husbandry Development Co., Ltd., Jiangmen 529000, China; waterfish8202@163.com

**Keywords:** tributyrin, broiler, growth performance, microbiota, biochemical parameter

## Abstract

**Simple Summary:**

Various antibiotic products in poultry production are gradually being banned around the world due to the adverse problems including the antibiotic residues and antibiotic resistance. Tributyrin was a potential alternative to antibiotics. The results of the present study indicated that tributyrin could improve the growth performance by modulating blood biochemical indices and the cecal microflora composition of yellow-feathered broilers. To the best of our knowledge, few studies investigated the effects of tributyrin on intestinal microbiota and its relationship with growth performance in broilers. This will provide a scientific basis for the application of tributyrin in animal husbandry in this post-antibiotic era.

**Abstract:**

This study aimed to evaluate the effects of tributyrin on growth performance, biochemical indices and intestinal microbiota of yellow-feathered broilers. 360 one-day-old chicks were randomly allocated to three treatments with six replicates of 20 chicks each, including a normal control group (NC), an antibiotic group (PC), and a tributyrin (250 mg/kg) group (TB) for 63 days. The results showed that compared with the control, the feed conversion ratio (FCR) in the TB group decreased during the d22 to d42 (*p* < 0.05) and overall, the final weight and FCR of broilers tended to increase and decrease, respectively. Moreover, the TB group showed the highest creatine concentrations at the entire period (*p* < 0.05). TB treatment increased the *Bacteroidetes* relative abundance and decreased *Firmicutes*. Principal coordinates analysis yielded clear clustering of the three groups. Linear discriminant analysis effect size analysis found seven differentially abundant taxa in the TB group, including several members of *Bacteroidedetes*. The relative abundance of *Eisenbergiella*, *Phascolarctobacterium*, *Megasphaera* and *Intestinimonas* increased in tributyrin-treated broilers. Spearman correlation analysis identified a correlation between *Eisenbergiella* abundance and overall feed efficiency. These results demonstrated that tributyrin could improve the growth performance by modulating blood biochemical indices and the cecal microflora composition of broilers.

## 1. Introduction

Sub-therapeutic antibiotics have been extensively used in the poultry industry for nearly 70 years to improve growth performance and disease resistance [1]. However, due to the rapid emergence of antibiotic-resistant bacteria and its potential threat to human health, an antibiotic ban in animal husbandry has been widely implemented, especially in the European Union in 2006 [2] and China in 2020 [3]. Therefore, there is an urgent need to find safe alternatives with no residual effect in this post-antibiotic era.

Increasing evidence suggests that butyrate is a potential alternative to antibiotics accounted for the fact that it can act as the major energy source of the colon [4], reduce pathogen colonization [5,6], improve intestinal barrier function [7], and exert anti-inflammatory properties [8]. However, the unpleasant odor and the rapid metabolism of butyric acid have limited its application in the livestock and poultry industry [9]. To overcome these drawbacks, there is increasing interest in developing butyric acid derivatives. Tributyrin is a triglyceride containing three molecules of butyric acid [10]. Previous research has shown that tributyrin improved growth performance in *Eimeria maxima*-infected broilers [11]. Moreover, dietary supplementation with 1000 mg/kg tributyrin improved eggshell quality and follicle-stimulating hormone secretion in broiler breeders with different egg-laying rates [12]. Tributyrin has the ability to converse the pepsinogen into pepsin, thus increasing the activity of the proteolytic enzymes and protein digestion [13]. A previous study reported that tributyrin increased the growth performance of weaned piglets accompanied with improvement of lipid metabolism and the protein digestibility for decreasing blood urea [14], a negative indicator of lean gain and feed efficiency [15]. He, et al. [16] reported that diet supplementation with 0.1% tributyrin could efficiently attenuate abnormal lipid metabolism in intrauterine growth retardation piglets.

It is widely acknowledged that the gastrointestinal tract (GIT) of chickens harbors a diverse community of microorganisms, which could reportedly positively affect nutrition absorption, immune system development, and disease resistance [17]. In this regard, intestinal microbiota and their metabolic products could influence the productivity and health of chickens [18]. Interestingly, the avian GIT bacteria has been documented to be modulated by diets and feed additives, such as probiotics, prebiotics and organic acids [18,19]. In recent years, the accelerated development of high throughput sequencing technology has expanded our understanding of diet, gut microbiota, and host interaction. Patrone, et al. [20] reported that diets supplemented with 0.2% tributyrin could positively affect gut microbial communities and the performance of weaning piglets. Likewise, supplementation with 5 g/kg monobutyrin and tributyrin was found to attenuate excessive inflammation in high-fat diet-fed mice by modulating the intestinal microbiota and decreasing liver cholesterol concentrations [21]. Furthermore, the intestinal microbiota can modulate the host health by regulating the biochemical profile, which could be reflected by blood biochemical indices and intestinal metabolic profiles [22,23]. However, to the best of our knowledge, few studies investigated the effects of tributyrin on intestinal microbiota in poultry. Therefore, we hypothesized that dietary tributyrin supplementation improved the growth performance of broilers by modulating plasma biochemical indices and the cecal microflora composition. Accordingly, this study investigated the effect of dietary tributyrin on the growth performance, blood biochemical indices, and intestinal microbiota of yellow-feathered broilers, aiming to find the potential microbiota for better growth performance of broilers.

## 2. Materials and Methods

### 2.1. Experimental Design and Diets

A total of 360 one-day-old female yellow-feathered broiler chicks were randomly allocated to three treatment groups of 120. Each group had six replicates with 20 chicks per replicate. The birds in the control group (NC) were fed a basal diet, while the antibiotic group (PC) and the tributyrin group (TB) fed a basal diet supplemented with a combination of 250 mg/kg nosiheptide and 200 mg/kg oxytetracycline calciuma, and 250 mg/kg tributyrin, respectively. The basal diets (Table 1) were formulated to meet the nutritional requirements of broilers recommended by the Ministry of Agriculture of the People’s Republic of China (2004). The tributyrin products that contained 50% tributyrin and silica as a carrier were purchased by Guangdong Herbio Biotechnology Co., Ltd., Guangzhou, China. The experiment period lasted 63 days and was divided into a starter phase (1–21 d), grower phase (22–42 d) and finisher phase (43–63 d). During the grower phase, 20 chicks with one replicate were assigned to a single page (65 × 60 × 40 cm), and each replicate was kept in two cages of this size during the grower phase and finisher phase. Chicks were reared in an environmentally controlled room with fresh water and feed provided ad libitum. The room temperature was maintained at 32 °C to 34 °C for the first three days and then reduced by 3 °C to 5 °C per week to achieve a final temperature of 26 °C. The light was provided for 23 h for the first week with a reduction to 16 h thereafter. 

### 2.2. Growth Performance and Organ Indices

At the day 21, 42 and 63 after 12 h feed withdrawal, the body weight and the feed residues within each replicate was recorded, and the average daily gain, the average feed intake and feed conversion ratio of each replicate were calculated and adjusted for mortality, which was recorded daily.

After weighting at the day 21, 42 and 63, one chicken close to the average weight of each replicate was selected, weighed, collected blood samples and tissue, and slaughtered by exsanguination. The liver, bursa of Fabricius and spleen were collected and washed with cold sterile PBS to remove the attached impurity. The organ weights were recorded and the organ indices were expressed as the percentage of body weight. 

### 2.3. Analyses of Plasma Biochemical Indicators 

At the day 21, 42 and 63, the blood samples (5 mL) were collected into ethylene diamine tetraacetic acid (EDTA) tubes by wing-vein puncture. The plasma was then obtained by centrifuged of the whole blood at 3500× *g* for 10 min at 4 °C and stored at −80 °C until biochemical indices analysis. Plasma (1 mL) was analyzed for uric acid, lactate dehydrogenase, creatinine, alkaline phosphatase, cholesterol, triglyceride, high-density lipoprotein, low-density lipoprotein, alanine transaminase, aspartate transaminase, and γ-glutamyl transpeptidase using a Biochemical Analyzer Architect C8000 (Abbott, Chicago, IL, USA). These plasma biochemical indices were measured with corresponding reagent kits all from Fangcheng Biotechnology Co., Ltd. (Beijing, China).

### 2.4. Microbial Analysis

On day 63, five of the selected six chickens per group were used for cecal microbiota analysis. The cecal contents were aseptically collected and stored at −80 °C after being snap frozen in liquid nitrogen. Total genome DNA from cecal digesta was extracted using QIAamp (Tiangen, Beijing, China). DNA concentration and purity were detected by 1.5% agarose gel electrophoresis. The V3/V4 region of the 16S rRNA gene was amplified by using the primer 341F (5′-CCTACGGGNGGCWGCAG-3′) and primer 805R (5′-GACTACHVGGGTATCTAATCC-3′), and samples were sequenced on an Illumina MiSeq platform. Sequences were analyzed using the QIIME software package (QIIME development team, http://qiime.org/index.html, accessed on 16 October 2021) [24]. The raw data were filtered first and the clean sequences were joined and analyzed following the standard QIIME pipeline. High-quality sequences were clustered into operational taxonomic unit (OTU) at 97% similarity and the representative OTU sequences were assigned taxonomy based on the SILVA database [25]. Shared and unique OTUs among three groups were visualized using a Venn diagram. The alpha diversity was calculated respectively by package vegan and plotted with graphpad prism 7.0. For beta-diversity analysis, a principal coordinates analysis (PCoA) was conducted based on Bray-Curtis distance with statistical significance determined by permutational multivariate analysis of variance (PERMANOVA) using R package. The differences in the relative abundances of microbiota were analyzed by linear discriminant analysis effect size (LEfSe, http://huttenhower.sph.harvard.edu/lefse/, accessed on 16 October 2021) and STAMP with *t*-test [26]. The Spearman correlation between growth performance and the representative genus was plotted as a heatmap using the R package pheatmap.

### 2.5. Statistical Analysis

All data were analyzed using one-way analysis of variance (ANOVA) followed by Tukey-Kramer multiple comparisons test using GraphPad Prism 7.0 (GraphPad Software, San Diego, CA). The tabular data are presented as means, standard error of the mean, and probability values. Statistical significance was accepted at a *p* < 0.05 and tendency at *p* < 0.1.

## 3. Results

### 3.1. Effects of Tributyrin on Growth Performance 

The effects of dietary tributyrin on the growth performance of yellow-feathered broilers are shown in Table 2. The final weight in the PC group was significantly higher than in the NC group (*p* < 0.05). In the PC group, the average daily gain (ADG) tended to increase compared with the control from day 1–63 (*p* = 0.063). Compared with the NC, the TB group decreased the feed conversion ratio (FCR) during the growth phase (22–42 d) (*p* < 0.05) and overall (*p* = 0.068) and showed a tendency to increase the final weight (*p* = 0.089). There were no significant differences in ADG and average daily feed intake (ADFI) (*p* > 0.05).

### 3.2. Effects of Tributyrin on Organ Indices

As shown in Table 3, no significant differences in organ indices including bursa, spleen, and liver were observed at all phases among the three groups (*p* > 0.05). 

### 3.3. Effects of Tributyrin on Biochemical Indices

As shown in Table 4, compared with the NC group, dietary antibiotic supplementation increased the uric acid concentration on day 21, while dietary tributyrin supplementation decreased on day 63 (*p* < 0.05). Lactate dehydrogenase activity was the highest in the control broilers on day 42 (*p* < 0.05). Significantly higher creatine concentrations were observed in the TB group than in the NC and PC groups on day 21, 42, and 63 (*p* < 0.05). Broilers treated with antibiotics showed higher alanine transaminase (ALT) activities than the TB group broilers on day 63 and NC broilers on day 21 and day 63 (*p* < 0.05). There was no significant difference in alkaline phosphatase, cholesterol, triglyceride, high-density lipoprotein, low-density lipoprotein, aspartate transaminase (AST), and γ-glutamyl transpeptidase at all sampling days.

### 3.4. Effects of Tributyrin on Intestinal Microbiota

A total of 1347 OTUs were identified in the cecal samples obtained from yellow-feathered broilers. The number of shared OTUs among the three groups was 489, and the numbers of OTUs specific for NC, PC and TB groups were 272, 119 and 81, respectively (Figure 1a). The composition of the top 10 phyla and top 15 genera in the cecal contents of broilers is shown in Figure 1b,c. The predominant phyla were *Bacteroidetes*, *Firmicutes*, *Tenericute*, *Proteobacteria*, *Actinobacteria*, and *Verrucomicrobia*, contributing to 99% of the whole phyla. Compared with the NC and PC groups, broilers in the TB group had a higher relative abundance of *Bacteroidetes* and a lower abundance of *Firmicutes*. Dominant microbiota in cecal contents at the genus level consisted of *Bacteroides*, unidentified*_Ruminococcaceae*, *Barnesiella*, *Lactobacillus*, *Akkermansia*, *Faecalibacterium*, unidentified*_Lachnospiraceae*, and *Gallibacterium*.

Regarding alpha and beta diversity of cecal microbiota in yellow-feathered broilers, results are shown in Figure 2. For alpha diversity indices, including observed_species, PD_whole_tree, shannon, simpson, chao1 and ace, no significant differences were found among the three groups (Figure 2a). Furthermore, beta diversity showed differences between groups (Figure 2b, *p* < 0.05). Likewise, the principal coordinates analysis (PCoA) based on Bray-Curtis distance revealed clear clustering of the three groups, though there was no significant difference between the NC and PC groups. The first principal components (PC1) accounted for 40% of the total variance in gut microbiota composition and the second principal components (PC2) accounted for 19% of the variation. The PC1 axis of the PCoA clearly separated the bacterial communities between the NC and TB groups.

The effect size measurements (LEfSe) analysis identified 18 biomarkers with LDA scores greater than four. Furthermore, seven bacterial taxa were found to be enriched in the TB group, including *Bacteroidales* (order), *Bacteroidia* (class), *Bacteroidedetes* (phylum), *Bacteroides_barnesiae* (species), *Selenomonadales* (order), *Negatlvlcutes* (class), and *Lactobacillus_mucase* (species). Moreover, *Lactobacillus* (genus), *Lactobacillaceae* (family), *Lactobacillales* (order), *Bacilli* (class), *Lactobacillus_aviarius* (species) were enriched in the PC group, while *Ruminococcaceae* (family), *Clostridiales* (order), *Clostridia* (class), *Firmicutes* (phylum), unidentified*_Ruminacaccaceae* (genus), *Clostridium_papyrosolvens* (species) were enriched in the NC group (Figure 3).

Welch’s *t*-test was further applied to identify the microbial composition differences at the order and genus levels between two groups, including NC vs. TB and PC vs. TB (Figure 4). In tributyrin-treated broilers, an increase in the relative abundance of *Bacteroidales* at the order level and lower *Clostridiales* abundance was observed, while the relative abundance of *Eisenbergiella*, *Phascolarctobacterium*, *Megasphaera* and *Intestinimonas* was increased. Moreover, in the TB group, a lower relative abundance of unidentified*_Ruminococcaceae* was found than in the NC group. After tributyrin supplementation, the relative abundance of *Lachnoclostridium* and *Lactobacillus* were increased and decreased, respectively, compared with the PC group.

Finally, the results of Spearman’s correlation analysis between growth performance during day1 to 63 and the top 35 genera in the yellow-feathered broilers are presented in Figure 5. The average abundance of the genera unidentified*_Mollicutes* was positively correlated with BW and ADG, whereas the genera of *Megamonas*, *Sutterell* and *Eisenbergiella* were negatively correlated with ADG and BW, ADFI, and FCR, respectively.

## 4. Discussion

Dietary supplementation with tributyrin has been demonstrated to improve animal production performance and disease resistance in monogastrics [27,28,29] and ruminants [30]. Notably, the positive effect of tributyrin may be attributed to the production of butyrate, which can promote the proliferation and differentiation of intestinal mucosal cells and improve nutrient absorption and utilization [31,32]. However, the efficacy of tributyrin as a growth promoter remains a matter of debate. It has been reported that dietary tributyrin administration at 1840 mg/kg increased the ADG, FCR and beneficial bacterial population in Arbor Acres (AA) broiler chickens under an isocaloric feeding regime [33]. Moreover, diets with a 0.2% mixture of butyric acid glycerides (mono-, di- and tri-glycerides) increased live weight at slaughtering and FCR of Ross 308 female chickens [34]. In contrast, tributyrate glyceride at 500 ppm or 2000 ppm did not change the ADG or FCR in Ross 308 and Ross 708 broilers; however, it modulated lipid metabolism and reduced abdominal fat deposition [35]. In the present study, we found that dietary supplementation with tributyrin increased the final weight and decreased the FCR in the entire phase; however, the difference failed to reach statistical significance. This discrepancy could be due to many factors, including different additions, breed, age, and different experimental conditions.

Blood biochemical indices usually reflect the physiological and metabolic status of animals. Creatinine, urea, and uric acid are widely acknowledged as reliable indicators for kidney function, as they are nitrogenous and protein end products of a catabolic process that are generally excreted by the kidney [36]. Elevated concentration of these plasma indices indicate kidney damage while decreasing their concentration helps to reduce the development of kidney disease [37]. In this regard, tributyrin has previously been reported to reduce kidney damage in dairy cows under heat stress by decreasing the concentration of creatinine, aspartate aminotransferase, total bilirubin, and the inflammatory response of immune cells [38]. Our results showed that tributyrin decreased the creatinine during all phases and uric acid levels during the last phase, indicating that tributyrin may have a potent protective effect on renal dysfunction. The mechanisms underlying this protective effect remain largely unknown, warranting further studies. Notably, we found that the uric acid concentration in the TB group was decreased on day 63 due to tributyrin’s previously documented long-term ability to reduce uricase activity [39]. Moreover, uric acid is a water-soluble antioxidant that inhibits oxidative damage and protects cell membranes and DNA [40,41]. The decreased uric acid concentration in TB groups contributed to the alleviation of oxidative injury of broilers. It was reported that elevated serum lactate dehydrogenase (LDH) activity together with alanine transaminase (ALT) and aspartate transaminase (AST) could serve as indicators of liver damage [42]. A previous study showed that the activity of ALT and AST in Japanese quails supplemented with sodium butyrate had no significant effects at day 21 but significantly decreased at day 42 [7]. Consistently, the current study showed that treatment with tributyrin reduced the ALT activity at day 63 and LDH at day 42, suggesting that tributyrin seems to protect the broilers from liver injury during the late phase.

It is widely acknowledged that commensal gut microbiota plays a crucial role in maintaining host animal’s health and normal physiology [17]. Accumulating evidence indicates *Firmicutes* dominate the cecal microflora in chickens [43,44]. Nonetheless, our study showed that *Bacteroidetes* was the dominant phylum of cecal contents in the TB group, accounting for more than 80%, whereas *Firmicutes* accounted for less than 20%, which was consistent with the literature [45]. Furthermore, our LEfSe results showed that the *Bacteroides* phylum and its class *Bacteroidia*, order *Bacteroidales*, species *Bacteroides_barnesiae*, were potential biomarkers in the TB group. *Bacteroides* are anaerobic gram-negative rods that are considered the major bacteria involved in producing SCFA [46]. Given that *Bacteroides* possess large repertoires of carbohydrate utilization genes [47], it can be inferred that *Bacteroides* can improve the host’s nutrient digestion and absorption through carbohydrate metabolism in tributyrin-treated chickens. Interestingly, a previous study reported the association between beneficial bacteria *Bacteroides* in the avian gut and increased growth performance, body weight gain, breast muscle yield and low abdominal fat [48]. Further research has shown that *Bacteroides* were significantly more abundant in the high feed efficiency (low FCR) group [45], indicating the low FCR and high final weight may be due to the increased *Bacteroides* in the TB group.

The genera *Eisenbergiella* and *Lachnoclostridium* belonging to the *Lachnospiraceae* family, *Intestinimonas* and *Megasphaera*, have been reported to be important butyrate producers residing in the intestinal microbiota [49,50,51]. The high relative abundance of genera *Eisenbergiella*, *Lachnoclostridium*, *Megasphaera* and *Intestinimonas* observed in the present study indicated that tributyrin strongly helped stimulate the synthesis of butyrate, the preferred energy source for the intestinal epithelial cells [52]. Additionally, the *Lachnospiraceae* family can reportedly degrade starch and nonstarch polysaccharides to produce organic acids [53], thereby decreasing the expression of virulence factors from pathogens like *Salmonella* [54]. Furthermore, *Lachnospiraceae* has been documented to be consistently depleted in people with acute colitis, suggesting its beneficial effects on maintaining intestinal homeostasis [55]. However, a previous study showed high *Eisenbergiella* abundance in infected mice, suggesting its association with gut dysbiosis [56]. There is limited information about the biological function of this genus, especially in birds, therefore needing further explore. Spearman correlation analysis in our study showed that *Eisenbergiella* was negatively correlated with FCR, suggesting that the low FCR in the TB group may be attributed to the increased abundance of *Eisenbergiella*. Tributyrin treatment was also found to increase *Phascolarctobacterium* abundance. A high-fat diet is reportedly more likely to result in an increased abundance of *Phascolarctobacterium* [57]. *Phascolarctobacterium* is a propionate producer, and a decline in the abundance of this genus has been associated with colonic inflammation [58]. In our study, the increased abundance of this genus in the TB group seemed to improve the host immune response to a certain extent.

In conclusion, dietary tributyrin supplementation decreased the FCR during the growth phase and as well as overall. Furthermore, it showed a tendency to increase the final weight. Tributyrin increased the creatine concentrations at the entire period and changed the cecal microflora composition of yellow-feathered broilers on day 63. Furthermore, spearman correlation analysis in the current study showed that *Eisenbergiella* was negatively correlated with FCR, suggesting that the low FCR in the TB group might be attributed to the increased *Eisenbergiella* abundance. These findings provided a new insight into the application of tributyrin as a potential antibiotic alternative in broiler industry by modulating the intestinal microbiota composition. Further research should explore the evolution of the cecal microbiota in TB-supplemented broilers and confirm the relationship between the *Eisenbergiella* and FCR.

## Figures and Tables

**Figure 1 animals-11-03425-f001:**
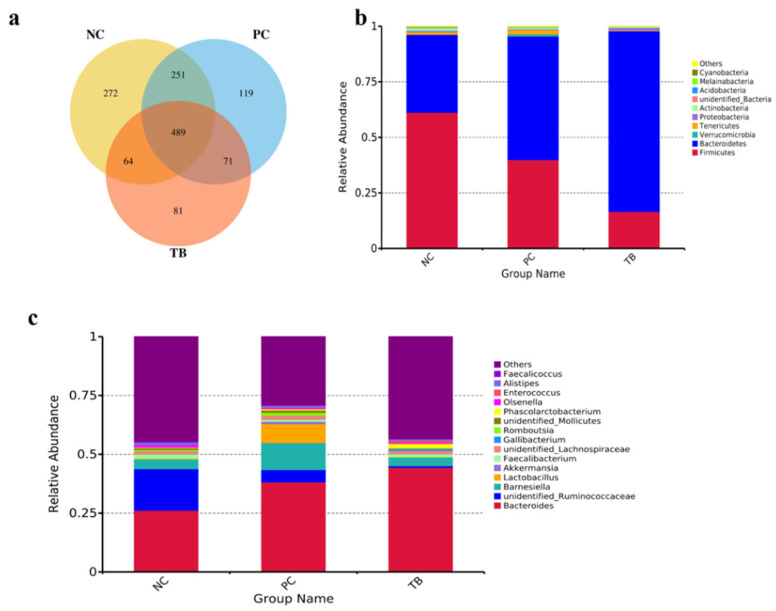
The Venn diagram and bacterial community composition of yellow-feathered broilers on day 63. (**a**) The Venn diagram of OTUs. Microbial composition in the cecum at the phylum (**b**) or genus (**c**) level. Taxa with relative abundance < 1% were combined into “other”. Abbreviations: NC, control group; PC, antibiotic group; TB, tributyrin group.

**Figure 2 animals-11-03425-f002:**
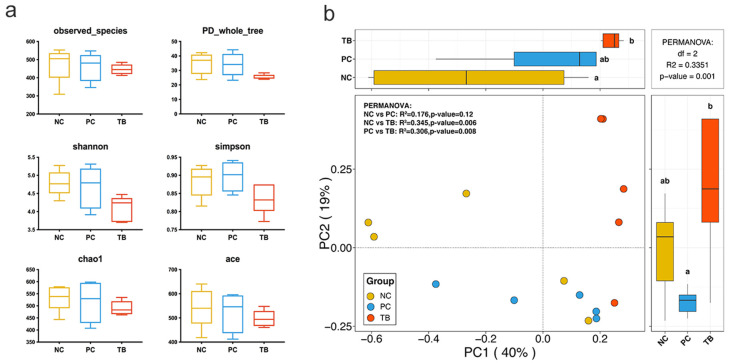
Diversity analyses of microbial communities among groups. (**a**) Alpha diversity analysis. (**b**) Principal coordinates analysis (PCoA) based on Bray-Curtis distance. Abbreviations: NC, control group; PC, antibiotic group; TB, tributyrin group.

**Figure 3 animals-11-03425-f003:**
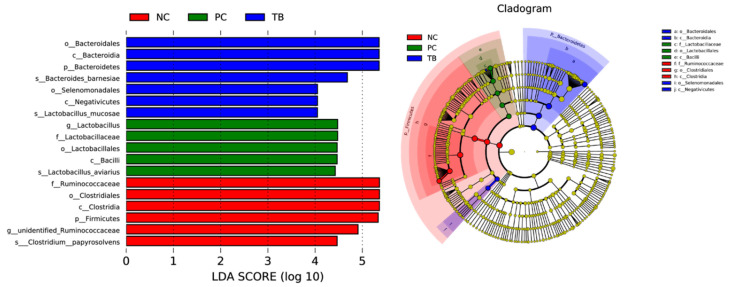
Linear discriminant analysis (LDA) effect size (LEfSe) analysis (*p* < 0.05, LDA > 4.0) of the intestinal microbes. The prefixes “p”, “c”, “o”, “f”, “g” and “s” represent the annotated level of phylum, class, order, family, genus and species. Abbreviations: NC, control group; PC, antibiotic group; TB, tributyrin group.

**Figure 4 animals-11-03425-f004:**
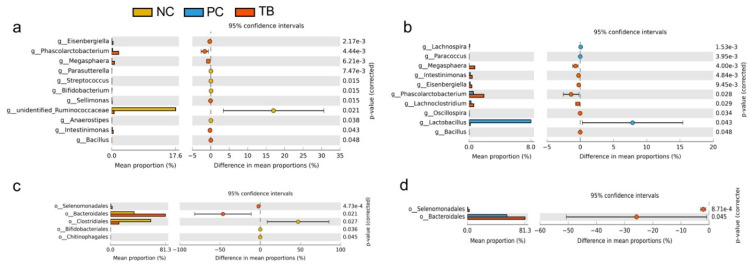
T-test analysis for the significant changes of differential microbiota at genus and order level in the cecal contents of yellow-feathered broilers. (**a**) NC vs. TB at genus level. (**b**) PC vs. TB at genus level. (**c**) NC vs. TB at order level. (**d**) PC vs. TB at order level. Abbreviations: NC, control group; PC, antibiotic group; TB, tributyrin group; g, genus; o, order.

**Figure 5 animals-11-03425-f005:**
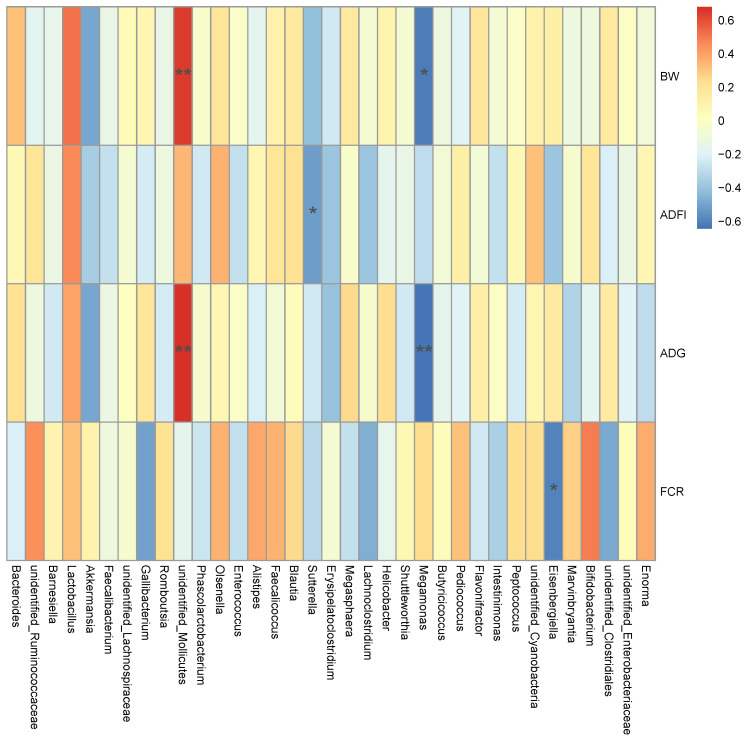
The Spearman correlation analysis of cecal microbial species at genus level with growth performance of yellow-feathered broilers. Spearman correlation coefficients are represented by color ranging from red, positive correlation (0.6), to blue, negative correlation (−0.6). * indicates statistically significant difference (*p* < 0.05). ** indicates extremely significant difference (*p* < 0.01). Abbreviations: BW, body weight; ADFI, average daily feed intake; ADG, average daily gain; FCR, feed conversion ratio.

**Table 1 animals-11-03425-t001:** The ingredient and nutrient level of the basal diet.

	Day 1–21	Day 22–42	Day 43–63
Feed ingredients (%)			
Corn	61.00	63.26	65.52
Soybean meal	32.00	28.00	24.00
Corn gluten meal	1.50	2.00	3.00
Soybean oil	1.40	2.50	3.50
Limestone	1.41	1.41	1.35
Dicalcium phosphate	1.33	1.33	1.33
*DL*-Methionine (99%)	0.18	0.15	0.12
*L*-Lysine-HCl (78%)	0.18	0.18	0.18
Wheat middling	0.11	0.17	0.00
Vitamin-mineral premix ^a^	1.00	1.00	1.00
Calculated nutrient composition ^b^			
ME (MJ/kg)	12.12	12.54	12.96
CP (%)	19.91	18.63	17.60
Lys (%)	1.09	1.00	0.92
Met (%)	0.51	0.46	0.42
Ca (%)	0.87	0.88	0.84
Available Phosphorus (%)	0.42	0.40	0.38

^a^ The premix provided the following per kg of diet: VA, 6000 IU; VD3, 2000 IU; VE, 30 mg; VK3, 2 mg; VB1, 3 mg; VB2, 5 mg; pantothenic acid, 800 mg; choline chloride, 1500 mg; nicotinic acid, 30 mg; pyridoxine, 3 mg; folic acid, 500 mg; biotin, 0.2 mg; VB12, 1 mg; Fe, 100 mg; Cu, 8 mg; Mn, 100 mg; Zn, 100 mg; I, 0.42 mg; Se, 0.3 mg. Sodium Chloride, 3000 mg; ^b^ Values were calculated from data provided by Feed Database in China (2004).

**Table 2 animals-11-03425-t002:** Effects of tributyrin on growth performance of yellow-feathered broilers.

Items		Treatment		SEM	*p*-Value
	NC	PC	TB		
Initial BW, g	36.57	36.95	37.41	0.50	0.519
Final BW, g	1581.96 ^b^	1624.97 ^a^	1609.84	13.56	0.103
ADG, g/d					
1–21 d	16.98	17.73	17.55	0.28	0.185
22–42 d	34.03	34.40	35.94	0.59	0.524
43–63 d	26.23	27.10	26.40	0.57	0.286
1–63 d	25.74	26.41	26.24	0.35	0.064
ADFI, g/d					
1–21 d	27.26	26.96	27.50	0.44	0.185
22–42 d	72.39	72.98	70.99	0.85	0.542
43–63 d	79.79	82.56	80.05	0.95	0.798
1–63 d	59.81	60.83	59.51	0.56	0.363
FCR					
1–21 d	1.61	1.52	1.57	0.02	0.061
22–42 d	2.13 ^b^	2.12	2.04 ^a^	0.03	0.349
43–63 d	3.04	3.05	3.03	0.06	0.617
1–63 d	2.32	2.30	2.27	0.02	0.341

Results are presented as mean, SEM, *p* value (*n* = 6). ^a,b^ In a row, means assigned different lowercase letters are significantly different (*p* < 0.05). Abbreviations: NC, control group; PC, antibiotic group; TB, tributyrin group; SEM, standard error of mean. BW, body weight; ADG, average daily gain; ADFI, average daily feed intake; FCR, feed conversion ratio.

**Table 3 animals-11-03425-t003:** Effects of tributyrin on organ indices of yellow-feathered broilers.

Items (% BW)		Treatment		SEM	*p*-Value
	NC	PC	TB		
Bursa index					
21 d	0.26	0.25	0.24	0.007	0.5053
42 d	0.18	0.23	0.21	0.008	0.0748
63 d	0.11	0.13	0.11	0.005	0.1552
Spleen index					
21 d	0.17	0.19	0.18	0.006	0.7373
42 d	0.18	0.17	0.17	0.005	0.6141
63 d	0.16	0.17	0.14	0.007	0.1878
Liver index					
21 d	3.53	3.48	3.87	0.083	0.1168
42 d	2.46	2.20	2.41	0.062	0.2286
63 d	1.66	1.59	1.73	0.039	0.3232

Abbreviations: NC, control group; PC, antibiotic group; TB, tributyrin group; SEM, standard error of mean (*n* = 6).

**Table 4 animals-11-03425-t004:** Effects of tributyrin on biochemical indices of yellow-feathered broilers.

Items		Treatment		SEM	*p*-Value
	NC	PC	TB		
Uric acid (μmol/L)					
21 d	181.50 ^b^	269.67 ^a^	219.67 ^ab^	12.02	0.0034
42 d	211.33	226.16	177.16	9.86	0.1102
63 d	257.50 ^a^	241.66 ^ab^	212.33 ^b^	7.39	0.0290
Lactate dehydrogenase (U/L)					
21 d	551.00	440.33	539.83	23.91	0.1105
42 d	733.83 ^a^	524.16 ^b^	461.66 ^b^	34.54	0.0003
63 d	539.83	540.83	440.00	20.24	0.0548
Creatinine (μmol/L)					
21 d	5.50 ^a^	6.00 ^a^	4.50 ^b^	0.19	0.0014
42 d	5.16 ^a^	4.83 ^a^	3.33 ^b^	0.28	0.0087
63 d	5.00 ^a^	4.83 ^a^	3.50 ^b^	0.22	0.0021
Alkaline phosphatase (U/L)					
21 d	1182.16	722.33	1312.50	118.67	0.0967
42 d	318.33	187.00	207.00	39.64	0.3675
63 d	64.83	78.33	79.66	8.80	0.7710
Cholesterol (mmol/L)					
21 d	3.53	3.48	3.70	0.12	0.7758
42 d	3.98	3.64	3.70	0.13	0.5541
63 d	3.56	3.10	3.55	0.09	0.0752
Triacylglycerols (mmol/L)					
21 d	0.44	0.50	0.62	0.03	0.0191
42 d	0.58	0.53	0.52	0.03	0.6933
63 d	0.51	0.53	0.59	0.02	0.4078
High density lipoprotein (mmol/L)					
21 d	2.46	2.28	2.44	0.07	0.5421
42 d	2.58	2.34	2.52	0.09	0.5494
63 d	2.33	2.03	2.24	0.06	0.1862
Low density lipoprotein (mmol/L)					
21 d	0.73	0.82	0.72	0.03	0.5318
42 d	0.66	0.72	0.63	0.03	0.4990
63 d	0.75	0.64	0.79	0.04	0.3524
Alanine aminotransferase (U/L)					
21 d	2.00 ^b^	3.16 ^a^	2.66 ^ab^	0.18	0.0216
42 d	1.83	2.00	1.83	0.13	0.8644
63 d	2.00 ^b^	3.16 ^a^	2.16 ^b^	0.1661	0.0019
Aspartate aminotransferase (U/L)					
21 d	201.50	222.66	212.16	4.88	0.2180
42 d	196.16	218.16	187.83	7.11	0.2052
63 d	214.00	210.66	221.83	6.84	0.8111
γ-glutamyl transpeptidase (U/L)					
21 d	27.66	26.83	23.50	1.30	0.4096
42 d	28.83	22.66	26.16	1.29	0.1488
63 d	24.66	26.66	23.33	1.23	0.5671

Results are presented as mean, SEM and *p* value (*n* = 6). ^a,b^ In a row, means assigned different lowercase letters are significantly different (*p* < 0.05). Abbreviations: NC, control group; PC, antibiotic group; TB, tributyrin group; SEM, standard error of mean.

## Data Availability

The data presented in this study are available on request from the corresponding author.

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
