# Peer review of "Effects of Dietary Tributyrin on Growth Performance, Biochemical Indices, and Intestinal Microbiota of Yellow-Feathered Broilers"

_animals, 2021, doi:10.3390/ani11123425_

Round 1

Reviewer 1 Report

FIgure 7 may be presented better, change the position of the graph

Author Response

November 20, 2021

Many thanks for your helpful evaluation and comments on our paper entitled " Effects of Dietary Tributyrin on Growth Performance, Biochemical Indices, and Intestinal Microbiota of Yellow-Feathered Broilers". We have studied comments carefully and tried to answer each of the questions to the best of our understanding.The manuscript has been revised according to your suggestion and all changes could be tracked with the Word file of the main manuscript.Our point-by- point responses to the comments are presented below. We sincerely hope this manuscript will meet the requirements of animals.

We hope that the revision is satisfactory and look forward to hearing from you soon if further revision is required.

Best wishes,

Li Gong

Point-to-point answers to reviewers’ comments:

-----------------

Reviewer 1#

Point 1

FIgure 7 may be presented better, change the position of the graph.

Response: Thanks for your professional suggestion and we have changed the position of the figure 7 (figure 4 in the revised manuscript).

Reviewer 2 Report

Reviewer Comments

The paper entitled “Effect of dietary tributyrin on growth performance, biochemical indices, and intestinal microbiota of yellow-feathered broilers” by L. Gong et al. addresses an interesting and important subject. The title of the article is correct and is consistent with its content. Moreover, the laboratory methodology is correct. However, the manuscript requires a major revision before it can be recommended for publication. The authors should especially address some concerns with the presentation of the materials and methods, and results section.

Specific comments:

Simple summary:

Line 16: please, clarify that the use of antibiotics in poultry production nowadays is a controlled fact. For instance, authors could say “Further, the traditional wide use of antibiotics in poultry production…”.

Abstract:

According to author instructions of Animals journal: “For research articles, abstracts should give a pertinent overview of the work. We strongly encourage authors to use the following style of structured abstracts, but without headings: (1) Background: Place the question addressed in a broad context and highlight the purpose of the study”. So please, add a short introduction in the abstract section.

Introduction:

Line 48: please, add references to the legislation on the prohibition of antibiotics in the European Union and China.

Line 54: please, add references.

Line 57: Eimeria maxima should be written in italic.

Line 58: if it is written that previous studies have shown this fact, more references must be added.

Line 58: please, add a connector for easier reading. For instance, “Moreover, dietary supplementation…”.

Line 65: please, add references.

Line 68: in this point of the introduction, two examples of other species are exposed, so a connector is needed.

Line 69: if the abbreviation has been indicated before (line 56), during all the manuscript “tributyrin” must be TB.

Line 74: please, add references of the studies.

Materials and methods:

This section should be re-written, it is very difficult to understand the real process of the experimental trial. Moreover, connectors should be added to easier reading.

Line 81: the code of the authorization should be added.

Line 87: the total number of animals for each group (n=120) should be added for better understanding.

Line 92: ad libitum must be written in italic.

Lines 92-93: the number of temperatures should be separated from the symbol, i.e., 26 °C.

Line 93: rewrite: “to achieve a final temperature of 26 °C”.

Lines 93-94: the sentence is not complete; a connector is needed. Moreover, why were the animals in continuous light exposure? To respect animal welfare, some hours of darkness are needed.

Line 95: Table 1: modify the structure of the table to be adapted to the available space.

Lines 96-103 (sample collection and treatment): this sector must be completed.

  • Line 98: an introductory sentence should be added, indicating all the samples collected in each of the sampling days.
  • Moreover, it should be explained the collection of all the performance data.

I consider that lines 98-103 should be in other section, this way materials and methods would be divided in:

  • Ethics Statement
  • Experimental Design and Diets
  • Sample collection: indicating all data recompilation (what, when and how samples were taken, and what, when and how performance data were obtained).
  • Analyses of plasma biochemical indicators: collection and analysis of blood samples.
  • Microbial Analysis: collection and analysis of fecal samples.
  • Statistical analysis.

Line 111: modify: On day 63, faeces for five chickens per group were taken…

  • How did you collect fecal samples?
  • Why fecal samples only were collected on day 63? What about the evolution of the fecal microbiota?

Lines 113-129: please, add references of microbial analysis.

Results:

In general, it is recommended to avoid repeating tabular data in the text (tautology) and concentrate on discussing and explaining the statistically significant differences.

Moreover, main results should be exposed in order, for example, the results regarding effects of tributyrin on growth performance and visceral index are very confusing.

Line 135 and line 143: please, unify in all the manuscript, or “indices” or “indexes”.

Line 142: replace “However” with “Regarding visceral indexes” or similar.

Line 153: Effects of tributyrin on biochemical indices: please, re-organize the results exposition and avoid repetition between text and figures.

Line 170: add a connector at the beginning of the sentence (“A total of” or similar).

Line 170: modify: samples obtained from

Line 170: if the abbreviation has been indicated before (line 119), during all the manuscript “operational taxonomic units” must be OTUs.

Line 171: please, add connector for easier reading.

Lines 176-177: “Compared with the NC and PC groups, broilers in the TB group 176 had a higher relative abundance of Bacteroidetes and a lower abundance of Firmicutes”,

  • Are there no differences between the other two experimental groups?

Lines 178-180: “Dominant microbiota in cecal contents at the genus level consisted of Bacteroides, uniden- 178 tified_Ruminococcaceae, Barnesiella, Lactobacillus, Akkermansia, Faecalibacterium, unidenti- 179 fied_Lachnospiraceae and Gallibacterium”,

  • Cecal content or fecal content?
  • In all the experimental groups or only in TB group?

Lines 182-185 (Figure 3): (a), (b) and (c) must be in bold type (according to authors instructions).

Line 186: an introductory sentence is needed for better understanding, for instance, “Regarding alpha and beta diversity of fecal microbiota in yellow-feathered broilers, results are shown in Figure 4” or similar.

Lines 187-189: please, re-order the sentence, “For alpha diversity indexes….”. Moreover, unify, or indices or indexes in all the manuscript.

Line 189: add connector, “Furthermore” or similar.

Line 189: “Beta diversity shows differences (Figure 4b, P < 0.05)”, please, add “between groups” at the end of the sentences. Moreover, all verbs must be in the same tense (past).

Lines 192-195: please, re-write for better understanding.

Lines 198-200 (Figure 4): (a), (b) and (c) must be in bold type (according to authors instructions).

Lines 201-209: please, indicate the experimental groups with their abbreviation for better understanding.

Line 214: which two groups? Please, indicate.

Lines 214, 219: “tributyrin” is “TB”.

Lines 214-218: TB group compared with which group)?

Lines 222-224 (Figure 6): (a), (b) and (c) must be in bold type (according to authors instructions).

Line 225: please, add a connector (“Finally” or similar).

Line 237, 239, 242 245, 246, etc.: “tributyrin” is “TB”.

Line 238: please, unify “indices”.

Lines 246-256: please, use the abbreviations of performance parameters.

Lines 257-280: this paragraph is confusing; it should be re-written and re-organized for easy reading.

Line 282: please, add reference.

Lines 282-286: how could you explain the reduction of Firmicutes in TB group? This paragraph should be re-organized to clarify.

Line 299: To which family do the genera Intestinimonas and Megasphaera belong?

Conclusion:

Lines 319-321: some information should be added at the conclusion of the manuscript.

Author Response

November 20, 2021

Many thanks for your helpful evaluation and comments on our paper entitled " Effects of Dietary Tributyrin on Growth Performance, Biochemical Indices, and Intestinal Microbiota of Yellow-Feathered Broilers". We have studied comments carefully and tried to answer each of the questions to the best of our understanding. The manuscript has been revised according to your suggestion and all changes could be tracked with the Word file of the main manuscript. Our point-by- point responses to the comments are presented below. We sincerely hope this manuscript will meet the requirements of animals.

We hope that the revision is satisfactory and look forward to hearing from you soon if further revision is required.

Best wishes,

Li Gong

Point-to-point answers to reviewers’ comments:

-----------------

Reviewer 2#

Reviewer Comments

The paper entitled “Effect of dietary tributyrin on growth performance, biochemical indices, and intestinal microbiota of yellow-feathered broilers” by L. Gong et al. addresses an interesting and important subject. The title of the article is correct and is consistent with its content. Moreover, the laboratory methodology is correct. However, the manuscript requires a major revision before it can be recommended for publication. The authors should especially address some concerns with the presentation of the materials and methods, and results section.

Specific comments:

Question1:

Simple summary:

Line 16: please, clarify that the use of antibiotics in poultry production nowadays is a controlled fact. For instance, authors could say “Further, the traditional wide use of antibiotics in poultry production...”.

Response: Many thanks for your careful and professional comment on our article. We sincerely apologize to the reviewer for the confusion. We have reworded the sentences, on page 1, line 14-16

Question2:

Abstract:

According to author instructions of Animals journal: “For research articles, abstracts should give a pertinent overview of the work. We strongly encourage authors to use the following style of structured abstracts, but without headings: (1) Background: Place the question addressed in a broad context and highlight the purpose of the study”. So please, add a short introduction in the abstract section.

Response: Many thanks for the reviewer’s valuable suggestion, but we are sorry for not adding a short introduction in the abstract section as suggested. Indeed, the abstract is limited within 200 words, and our abstract just reached 200 words, so we have to write only the purpose of the study in the beginning of the abstract. In our next manuscript, we have corrected it as suggested, and we wish it can also be reviewed by you. Thanks again for your professional comments.

Question3-4:

Introduction:

Line 48: please, add references to the legislation on the prohibition of antibiotics in the European Union and China.

Line 54: please, add references.

Response:  Thank you for your professional suggestions. we have added the related reference, on page 2, lines 44 and 50.

 Question 5-7:

Line 57: Eimeria maxima should be written in italic.

Line 58: if it is written that previous studies have shown this fact, more references must be added.

Line 58: please, add a connector for easier reading. For instance, “Moreover, dietary supplementation...”.

Response: Many thanks for pointing out our style and grammatical mistake, and we have corrected them in the revised manuscript as suggested, on page 2, lines 53 and 54.

Question 8-9:

Line 65: please, add references.

Line 74: please, add references of the studies.

Response: Thank you for your professional advice. Appropriate reference has been added, on page 2, lines 67. For the line 74, actually, to the best of our knowledge, there were no studies investigated the effects of tributyrin on intestinal microbiota in poultry.

Question 10:

Line 68: in this point of the introduction, two examples of other species are exposed, so a connector is needed.

Response: Thank you. the connector has been added, page 2, line 72.

Question 11:

Line 69: if the abbreviation has been indicated before (line 56), during all the manuscript “tributyrin” must be TB.

Response: Thank you for pointing out our errors. We have omitted the abbreviation of tributyrin throughout the manuscript.

Question 12:

Materials and methods:

This section should be re-written, it is very difficult to understand the real process of the experimental trial. Moreover, connectors should be added to easier reading.

Response: Thank you for your professional advice. we have re-written this section in the revised manuscript and the connectors also has been added for easier reading.

Question 13-14:

Line 81: the code of the authorization should be added.

Line 87: the total number of animals for each group (n=120) should be added for better understanding.
Response: Thank you. The code of the authorization and the total number of animals for each group has been added in the revised manuscript, page 3, line 89 and 92.

Question 15-16:

Line 92: ad libitum must be written in italic.

Lines 92-93: the number of temperatures should be separated from the symbol, i.e., 26 °C. Line 93: rewrite: “to achieve a final temperature of 26 °C”.

Response:  Many thanks for pointing out our style and grammatical mistake, and we have corrected them in the revised manuscript as suggested, on page 3, lines 105-106.

Question 17:

Lines 93-94: the sentence is not complete; a connector is needed. Moreover, why were the animals in continuous light exposure? To respect animal welfare, some hours of darkness are needed.

Response: Thank you for your professional comment. The connector has been added and the light time has been revised (page3, line 106-107).

Question 18:

Line 95: Table 1: modify the structure of the table to be adapted to the available space.

Response: Thanks. we have stated the composition right below the ingredients, line 108.

Question 19-22:

Lines 96-103 (sample collection and treatment): this sector must be completed.

-  Line 98: an introductory sentence should be added, indicating all the samples collected

in each of the sampling days.

-  Moreover, it should be explained the collection of all the performance data.

I consider that lines 98-103 should be in other section, this way materials and methods would be divided in:

- Ethics Statement
- Experimental Design and Diets
- Sample collection: indicating all data recompilation (what, when and how samples were taken, and what, when and how performance data were obtained).
- Analyses of plasma biochemical indicators: collection and analysis of blood samples. - Microbial Analysis: collection and analysis of fecal samples.
- Statistical analysis.

Response: We are grateful for your careful work and we have modified them as suggested in the revised manuscript, lines 114-160.

Question 23:

Line 111: modify: On day 63, faeces for five chickens per group were taken...
- How did you collect fecal samples?
- Why fecal samples only were collected on day 63? What about the evolution of the fecal microbiota?

Response: We gratefully appreciate the reviewer’s professional review work on our article. On day 63, five of the selected six chickens per group were used for cecal microbiota analysis. The cecal contents were aseptically collected and stored at -80 °C after being snap frozen in liquid nitrogen, on page 4, lines 135-137.

Indeed, it will be more convincing if the cecal microbiota is collected and analysis on day 21, 42 and 63 and it may better reveal the evolution of cecal microbiota of yellow feathered broilers. However, in the present study, via detecting cecal microbiota on day 63 and the final growth performance in broilers, we mainly focused on the interactions of the final growth performance and intestinal microbiota. We think that our experiment design may not be optimal, but it should be sufficient to draw the expected conclusion in yellow-feathered broilers. We deeply appreciate the reviewer's comment, and we would very much like to consider this advice in our future work.

Question 24:
Lines 113-129: please, add references of microbial analysis.

Response: Thanks for your valuable suggestions and we have added the related references, page 4, lines 142,146 and 153.

Question 25:

Results:

In general, it is recommended to avoid repeating tabular data in the text (tautology) and concentrate on discussing and explaining the statistically significant differences.

Moreover, main results should be exposed in order, for example, the results regarding effects of tributyrin on growth performance and visceral index are very confusing.

Response: We are sorry for the confusion. we have divided the related results into two paragraph, page 4-5, line 162 and 176.

Question 26:

Line 135 and line 143: please, unify in all the manuscript, or “indices” or “indexes”.

Response: Thanks for your careful work. We have unified the indices in all the manuscript.

Question 27:

Line 142: replace “However” with “Regarding visceral indexes” or similar.

Response: Thanks for your valuable suggestion. We have divided the results of organ indices into a separated paragraph, page 5, line 176.

Question 28:

Line 153: Effects of tributyrin on biochemical indices: please, re-organize the results exposition and avoid repetition between text and figures.

Response: Thank you for your professional comments. We have tabulated the results of the biochemical indicators in the tables.

Question 29:

Line 170: add a connector at the beginning of the sentence (“A total of” or similar).

Response: Thanks. we have added the connector as suggested, page 6, line 198.

Question 30-32:

Line 170: modify: samples obtained from...

Line 170: if the abbreviation has been indicated before (line 119), during all the manuscript “operational taxonomic units” must be OTUs.

Line 171: please, add connector for easier reading.

Response:  Thanks. we have corrected it as suggested, on page 6, lines 198- 199.

Question 33:

Lines 176-177: “Compared with the NC and PC groups, broilers in the TB group 176 had a higher relative abundance of Bacteroidetes and a lower abundance of Firmicutes”,

- Are there no differences between the other two experimental groups?

Response: Thanks. broilers in the TB group had a higher relative abundance of Bacteroidetes and a lower abundance of Firmicutes, compared with the NC (P < 0.05) and PC groups (P < 0.1). There was no difference between NC and PC.

Question 34:

Lines 178-180: “Dominant microbiota in cecal contents at the genus level consisted of Bacteroides, uniden- 178 tified_Ruminococcaceae, Barnesiella, Lactobacillus, Akkermansia, Faecalibacterium, unidenti- 179 fied_Lachnospiraceae and Gallibacterium”,

- Cecal content or fecal content?
- In all the experimental groups or only in TB group?

Response:  We are sorry for the confusion. We collected the cecal content of broilers for 16s RNA analysis. The results stated above are in all the experimental groups.

Question 35:

Lines 182-185 (Figure 3): (a), (b) and (c) must be in bold type (according to authors instructions).

Response: Thanks. We have corrected it, page 7, lines 211-212.

Question 36-37:

Line 186: an introductory sentence is needed for better understanding, for instance, “Regarding alpha and beta diversity of fecal microbiota in yellow-feathered broilers, results are shown in Figure 4” or similar.

Lines 187-189: please, re-order the sentence, “For alpha diversity indexes....”. Moreover, unify, or indices or indexes in all the manuscript.

Response: Many thanks for your professional advice. we have corrected it as suggested, lines 214-217.

Question 38:

Line 189: add connector, “Furthermore” or similar.

Response: Thanks. We have added the connector, on line 217.

Question 39:
Line 189: “Beta diversity shows differences (Figure 4b, P < 0.05)”, please, add “between groups” at the end of the sentences. Moreover, all verbs must be in the same tense (past).

Response: Thank you for pointing out our grammatical mistake and we have corrected it, on page 7, lines 217-218.

Question 40:

Lines 192-195: please, re-write for better understanding.

Response: We sincerely apologize to the reviewer for the confusion, and we have rewriting the sentences, on page 7, lines 220-223.

Question 41:

Lines 198-200 (Figure 4): (a), (b) and (c) must be in bold type (according to authors instructions).

Response: Thank you. We have corrected it, lines 226-227.

Question 42:

Lines 201-209: please, indicate the experimental groups with their abbreviation for better understanding.

Response: Thanks for your valuable suggestions. We have indicated the experimental groups with their abbreviation, lines 229-237.

Question 43:

Line 214: which two groups? Please, indicate.

Response: NC vs. TB and PC vs. TB. We have added it, line 242.

Question 44:

Lines 214, 219: “tributyrin” is “TB”.

Response: Thanks. We have omitted the abbreviations of tributyrin.

Question 45:

Lines 214-218: TB group compared with which group)?

Response:  We are sorry for the confusion. TB group compared with the NC and PC group.

Question 46:

Lines 222-224 (Figure 6): (a), (b) and (c) must be in bold type (according to authors instructions).

Response: Thanks. We have corrected it as suggested, line 252.

Question 47:

Line 225: please, add a connector (“Finally” or similar).

Response: Thanks. We have added the connector as suggested, page 9, line 254.

Question 48:

 Line 237, 239, 242 245, 246, etc.: “tributyrin” is “TB”.

Response: Thanks. We have omitted the abbreviations of tributyrin.

Question 49:

Line 238: please, unify “indices”.

Response: Thanks. we have removed this sentence.

Question 50:

Lines 246-256: please, use the abbreviations of performance parameters.

Response: Many thanks. We have revised the abbreviations of performance parameters, on page 10, lines 272-276.

Question 51:

Lines 257-280: this paragraph is confusing; it should be re-written and re-organized for easy reading.

Response: We apologize to you for the confusion. We have reconstructed the paragraph structure for easy reading, page 10-11, lines 283-306.

Question 52:

Line 282: please, add reference.

Response: Thanks for your professional advice and we have added the related reference, line 308.

Question 53:

Lines 282-286: how could you explain the reduction of Firmicutes in TB group? This paragraph should be re-organized to clarify.

Response:  Thanks for your professional comments. It was observed that high fat diet increased the proportion of Firmicutes to Bacteroidetes, while the increased concentration of dietary fiber might decrease the ratio. Furthermore, the reduced Firmicutes populations were positively related to the reduction in fat deposition.

  1. Peng, J.G. Yin, X.L. Liu, B.Y. Jia, Z.G. Chang, H.J. Lu, N.Jiang, Q.J. Chen First insights into the microbial diversity in the omasum and reticulum of bovine using Illumina sequencing

Journal of Applied Genetics, 56 (2015), pp. 393-401] Simpson, H.L.; Campbell, B.J. Review article: Dietary fiber–microbiota interactions. Aliment. Pharmacol. Ther. 2015, 42, 158–179.

Question 54:

Line 299: To which family do the genera Intestinimonas and Megasphaera belong?

Response: Thanks. The genera Intestinimonas belongs to the family Ruminococcaceae and the genera Megasphaera is assigned to a family Veillonellaceae.

Question 55:

Conclusion:
Lines 319-321: some information should be added at the conclusion of the manuscript.

Response: Thanks for your professional comments. The conclusion has been reworded, page 12, lines 345-355.

Reviewer 3 Report

To Authors

The manuscript is about an interesting and needed topic, and importantly, it can also have a practical aspect. Unfortunately, the work lacks a hypothesis and a well-defined research goal. This is a big drawback. The reader must guess for himself what the Authors wanted to investigate, evaluate, check. I believe that a scientific work should clearly formulate a hypothesis and research goal, which greatly facilitates the understanding of the Authors' way of thinking and enriches the publication a lot. I am asking for a correct hypothesis and a clearly formulated aim of the research.

Author Response

November 20, 2021

Many thanks for your helpful evaluation and comments on our paper entitled " Effects of Dietary Tributyrin on Growth Performance, Biochemical Indices, and Intestinal Microbiota of Yellow-Feathered Broilers". We have studied comments carefully and tried to answer each of the questions to the best of our understanding. The manuscript has been revised according to your suggestion and all changes could be tracked with the Word file of the main manuscript. Our point-by- point responses to the comments are presented below. We sincerely hope this manuscript will meet the requirements of animals.

We hope that the revision is satisfactory and look forward to hearing from you soon if further revision is required.

Best wishes,

Li Gong

Point-to-point answers to reviewers’ comments:

-----------------

Reviewer 3#

To Authors

The manuscript is about an interesting and needed topic, and importantly, it can also have a practical aspect. Unfortunately, the work lacks a hypothesis and a well-defined research goal. This is a big drawback. The reader must guess for himself what the Authors wanted to investigate, evaluate, check. I believe that a scientific work should clearly formulate a hypothesis and research goal, which greatly facilitates the understanding of the Authors' way of thinking and enriches the publication a lot. I am asking for a correct hypothesis and a clearly formulated aim of the research.

Question1:

Line 14-15: Please define the content differently to describe the thought the authors wish to convey. This sentence is too vague.

Response: Many thanks for your careful and professional comment on our article. We sincerely apologize to the reviewer for the confusion. We have reworded the sentences, on page 1, line 14-15.

Question2:

Line 40-86: In the Introduction section, the authors describe the advantages of using tributyrin in the context of replacing atibiotic growth stimulants with it. However, please also justify in this part what was the purposefulness of determining such and not other blood biochemical indicators and other parameters included in the study. This is imprecise and unclear. What was expected after identification them, what response? What is the possible link between these indicators and the expected results? What is the link with the effects of rearing? This needs to be clearly explained. As for the results and conclusions. Here's another problem. I did not find any research hypothesis in the work. Moreover, the purpose of the work is too laconic. Please clearly state the hypothesis that was verified in this study.

Response: Many thanks for your professional comments. We have reworded this section as suggested, page 1-2, lines 40-84.

Question3:

Line 43-44: I propose to quote other recent publications on the use of probiotics in the rearing of broiler chickens, which showed an improvement in rearing indicators, blood biochemical indicators, as well as intestinal morphometry and microbome composition. I suggest you analyze and quote:

  1. Abramowicz K, Krauze M, Ognik K. 2020. Use of Bacillus subtilis PB6 enriched with choline to improve growth performance, immune status, histological parameters and intestinal microbiota of broiler chickens. Anim Prod Sci. 60(5):625–634.
  2. Park I, Zimmerman NP, Smith AH, Rehberger TG, Lillehoj EP and Lillehoj HS (2020) Dietary supplementation with Bacillus subtilis direct-fed microbials alters chicken intestinal metabolite levels. Front. Vet. Sci. 7:123.

Response: Many thanks for recommending the related publications with good writing and we learned a lot from these two papers, so we have analyzed and quoted in the revised manuscript, on page 2, line 67,69 and 77.

Question4:

Line 60-75: Why is the research hypothesis missing from the publication? This is a very important element that guides the entire research process. Moreover, the purpose of the work is not precise. Under what angle were these studies conducted. Please complete it and clarify it.

Response: We are sincerely for the professional comments and we have clarified the hypothesis and purpose of this work in the revised manuscript, on page 2, lines 79-84.

We hypothesized that dietary tributyrin supplementation improved the growth performance of broilers by modulating plasma biochemical indices and the cecal microflora composition. Accordingly, this study investigated the effect of dietary tributyrin on the growth performance, blood biochemical indices, and intestinal microbiota of yellow-feathered broilers, aiming to find the potential microbiota for better growth performance of broilers.

Question5:

Line 86: How was the tributyrin dose determined? What was this preparation, what were its characteristics and origin? There is no mention of this. Please complete this and describe it exactly in the review, including producer.

Response: Thanks for your valuable suggestions. We have added the related information of tributyrin in the revised manuscript, page 3, lines 98-100.

The tributyrin products that contained 50% tributyrin and silica as a carrier were purchased by Guangdong Herbio Biotechnology Co.Ltd., Guangzhou, China.

Question6:

Line 90-93: Based on what guidelines were the zootechnical conditions for the rearing of chickens established? Please state it in the text.

Response: I am sorry for missing this information and we have added it, on page 3, lines 101-107.

During the grower phase, 20 chicks with one replicate were assigned to a single page (65 × 60 × 40 cm), and each replicate was kept in two cages of this size during the grower phase and finisher phase. Chicks were reared in an environmentally controlled room with fresh water and feed provided ad libitum. The room temperature was maintained at 32 °C to 34 °C for the first three days and then reduced by 3 °C to 5 °C per week to achieve a final temperature at of 26 °C. The light was provided for 23 h for the first week with a reduction to 16h continuous lighting was provided throughout thereafter the whole feeding period.

Question7:

Table 1: Header: "Nutrient composition" - Are these calculated or marked values? Please specify it in the table. If these are calculated values, please provide the source under the table on the basis of which the data is calculated.

Response: Thanks for your professional comments. The nutrient composition is calculated based on the data provided by Feed Database in China (2004). We have added the word “calculated” in the table of the ingredient and nutrient level of the basal diet, line 108.

Question8:

Line 97-98: How much blood was collected from each bird?

Response: Thanks. Blood (5 ml) was collected from each selected bird, page 4, line 125.

Question9:

Line 104-108: Please clarify in the text, with methods and possibly tests the blood biochemical indicators were determined? In addition, please explain to the reviewer in the review what amount of plasma was used to determine all these indicators?

Response: Thanks for your kind advice. We have added the kits information of blood biochemical indicators in the revised manuscript, page 4, lines 131-133. We used 1ml plasma to determine all these indicators.

Question10-11:

Figure 1: Line 145: Please convert the graphs to tabular data.

Figure 2: Line 164: Please tabulate the results of the biochemical indicators in the tables. The results are illegible in this form. Moreover, there are inaccuracies in the charts. In this context, when it comes to enzymes, we are talking about activity, not about level! Please correct this. The name triglycerides is an old name, now we are talking about triacylglycerols. In addition, we use well-known abbreviations for all indicators, not only for AST and ALT. Please apply them and standardize the data.

Response: Many thanks for your professional comments. We have converted the graphs to tabular data as suggested. For the grammatical and spelling errors, we have corrected them in the revised manuscript, on page 5-6, line 171, 188 and 193.

Question12:

Line 236-237: In the opinion of the reviewer it is not good start this part of the publication.

Response: We are grateful for your useful suggestions and we have removed the unsuitable beginnings of this section, page 10, line 266.

Question13:

Line 256: What's that strange statement? Please scientifically specify what the authors meant.

Response: We sincerely apologize to the reviewer for the confusion. We have reworded the sentences, on page 10, lines 283-284.

Question14-15:

Line 275: I guess the authors meant AST and ALT activity, right?
Line 277: The same error as line 271 and 275.

Response: We are sorry for the mistake and we have corrected it in the revised manuscript, on page 11, lines 302 and 304.

Question16:

Line 318-320: Please provide conclusions based on the hypothesis and purpose of the work. It would be worth, trying to provide information about the improvement of indicators, the improvement of which was correlated with growth performance? Also, please provide a practical conclusion from these tests.

Response: Thanks for your professional comments. The conclusion has been reworded as suggested in the revised manuscript, on page 12, lines 345-355.

Reviewer 4 Report

Line 41: I suggest the authors to remove such a statement. The fact that poultry meat consumption is increasing is not relevant at all for your study. Please, start the introduction stating about the problem your study can solve. In this case, animal gut health. I would suggest the following sequence: importance of gut health for digestion and absorption of nutrients > impairments in performance due to immune challenge > banning of antibiotics > the solution you propose as additive > studies or potential benefits of your additive for the host

Line 89: I believe the authors meant “phase, instead of “phase”

Line 93: Use “entire” instead of “whole”

Table 1: Please, do not split the table into one column for ingredients and other for composition. State the composition right below the ingredients. Were these diets formulated without any sodium source. It is impossible to meet Na requirements with cereals. Regarding the nutrients, did the authors formulate based on ME or AMEn? Total of SID amino acids? Total or available phosphorous? Include the Na content of feeds. Provided such details. Which were the requirements used as basis? Are the nutrients expressed s-is or dry matter basis? State about the sources of crystalline amino acids used and their purity (e.g., DL-Methionine, 98%) NRC, 1994 for broilers? If so, add to the description already provided that broilers were the specie chosen.

Line 99: “for further analysis” would be more proper.

Line 104: it would be better: “Plasma was analyzed for … using an Biochemical Analyzer Architect C8000…

Line 110: “per groups were used…”

Line 132: “tendency” would be a better word.

Results

Please, decide if you will use “compared with” or “compared to”.

Avoid using in this section words such as “Moreover”, “however”. Just state about effects.

Please, change the way the performance outcomes were detailed. Express them in table with SEM and p-values.

Include SEM in all your tables.

Check the tables for commas and dots. There are values expressed with commas.

Line 138-140. Split this sentence. It is too long.

Line 237-240: It could be summarized as follows. Probably, the reference number may change. “Tributyrin has shown to improve performance and disease resistance in monogastrics [16, 17, 19] and ruminants [18].”

Line 252-253: if it is not statistically different, it should not be mentioned the differences. Please, remove such statement. The authors should explore perhaps the feed composition as a factor which can contribute for the differences among studies. Cereals rich in NSPs increase viscosity of digesta and consequently modulate gut microbiota.   

Author Response

November 20, 2021

Many thanks for your helpful evaluation and comments on our paper entitled " Effects of Dietary Tributyrin on Growth Performance, Biochemical Indices, and Intestinal Microbiota of Yellow-Feathered Broilers". We have studied comments carefully and tried to answer each of the questions to the best of our understanding. The manuscript has been revised according to your suggestion and all changes could be tracked with the Word file of the main manuscript. Our point-by- point responses to the comments are presented below. We sincerely hope this manuscript will meet the requirements of animals.

We hope that the revision is satisfactory and look forward to hearing from you soon if further revision is required.

Best wishes,

Li Gong

Point-to-point answers to reviewers’ comments:

-----------------

Reviewer 4#

Question1:

Line 41: I suggest the authors to remove such a statement. The fact that poultry meat consumption is increasing is not relevant at all for your study. Please, start the introduction stating about the problem your study can solve. In this case, animal gut health. I would suggest the following sequence: importance of gut health for digestion and absorption of nutrients > impairments in performance due to immune challenge > banning of antibiotics > the solution you propose as additive > studies or potential benefits of your additive for the host.

Response: Many thanks for your careful work and professional comments. We have removed the irrelevant sentence of the first paragraph as you suggested. We are grateful for your suggested following sequence and we have corrected it in our next manuscript about the topic of tributyrin against E. coli challenge in broilers, and we wish it can also be reviewed by you. Thanks again for your careful work and valuable suggestions.

Question2:

Line 89: I believe the authors meant “phase, instead of “phase”

Response:  We are sorry for making spelling mistake and have corrected it in the revised manuscript, page 3, lines 100-101.

Question3:

Line 93: Use “entire” instead of “whole”

Response: Thanks for your kind advice and we have corrected in the revised manuscript throughout.

 Question4:

Table 1: Please, do not split the table into one column for ingredients and other for composition. State the composition right below the ingredients. Were these diets formulated without any sodium source. It is impossible to meet Na requirements with cereals. Regarding the nutrients, did the authors formulate based on ME or AMEn? Total of SID amino acids? Total or available phosphorous? Include the Na content of feeds. Provided such details. Which were the requirements used as basis? Are the nutrients expressed s-is or dry matter basis? State about the sources of crystalline amino acids used and their purity (e.g., DL-Methionine, 98%) NRC, 1994 for broilers? If so, add to the description already provided that broilers were the specie chosen.

Response:  Thank you for your professional comments. we have adjusted the table into one column as you suggested. We are sorry for missing the sodium chloride content(0.3%)in the table of ingredient and nutrient level of the basal diet and we have added it, page 3, line 108.

Regarding the nutrients, we formulated based on ME. Total amino acids. Available phosphorous.

 The basal diets were formulated to meet the nutritional requirements of broilers recommended by the Ministry of Agriculture of the People’s Republic of China (2004) and the values were calculated from data provided by Feed Database in China (2004). The nutrients are expressed in dry matter basis. The sources of crystalline amino acids used and their purity were stated in the revised manuscript, DL-Methionine (99%), L-Lysine-HCl(78%).

Question5:

Line 99: “for further analysis” would be more proper.

Response:  We are sorry for not clarifying properly and we have corrected it in the revised manuscript, page 4, line 127.

Question6-8:

Line 104: it would be better: “Plasma was analyzed for … using an Biochemical Analyzer Architect C8000…

Line 110: “per groups were used…”

Line 132: “tendency” would be a better word.

Response: Many thanks for your kind advice. We have corrected it as suggested, on page 4, line 127, 135 and 160.

Question9-10:

Results

Please, decide if you will use “compared with” or “compared to”.

Avoid using in this section words such as “Moreover”, “however”. Just state about effects.

Response: Thanks for your professional comments. We replaced “compared to” with “compared with” and removed some connectors in this section as suggested.

Question11-12:

Please, change the way the performance outcomes were detailed. Express them in table with SEM and p-values.

Include SEM in all your tables.

Response: Thank you for your professional comments. We have tabulated the results of the growth performance and biochemical indices in the tables and add the SEM, p value in all tables, lines 171, 179 and 193.

Question13:

Check the tables for commas and dots. There are values expressed with commas.

Response: Many thanks for pointing out the mistake. We have corrected it on page 5, line 179.

Question14:

Line 138-140. Split this sentence. It is too long.

Response: We sincerely apologize to the reviewer for the confusion. We have reworded the sentences, on page 5, lines 166-169.

Question15:

Line 237-240: It could be summarized as follows. Probably, the reference number may change. “Tributyrin has shown to improve performance and disease resistance in monogastrics [16, 17, 19] and ruminants [18].”

Response: Thank you for your valuable suggestion, we have made correction as suggested, page 10, lines 267-268.

Question16:

Line 252-253: if it is not statistically different, it should not be mentioned the differences. Please, remove such statement. The authors should explore perhaps the feed composition as a factor which can contribute for the differences among studies. Cereals rich in NSPs increase viscosity of digesta and consequently modulate gut microbiota.   

Response: Many thanks for your professional comment. In the current study, we have added the NSP enzyme to reduce the effects of NSPs on the viscosity of digesta and gut microbiota.

Indeed, we hypothesized that dietary tributyrin supplementation improved the growth performance of broilers by modulating plasma biochemical indices and the cecal microflora composition and we aimed to find the potential microbiota for better growth performance of broilers. Our results showed the relative abundance of Eisenbergiella was increased in tributyrin-treated broilers and the spearman correlation analysis identified a correlation between Eisenbergiella abundance and overall feed efficiency. The TB group decreased the feed conversion ratio (FCR) during the growth phase (P < 0.05) and overall (P = 0.0675). Further research should explore the evolution of the cecal microbiota in TB-supplemented broilers and confirm the relationship between the Eisenbergiella and FCR.

Round 2

Reviewer 2 Report

Dear authors,

Thank you very much for the improvement of the manuscript. After modifications, I consider it ready for publication.

Reviewer 4 Report

The authors considered the comments and the paper can be published